

# Metabolic response of *Scapharca subcrenata* to heat stress using GC/MS-based metabolomics

Yazhou Jiang[1], Haifeng Jiao[2], Peng Sun[1], Fei Yin[3] and Baojun Tang[1]

[1] East China Sea Fisheries Research Institute, Chinese Academy of Fishery Sciences, Shanghai, China
[2] Ningbo Academy of Ocean and Fishery, Ningbo, Zhejiang, China
[3] Key Laboratory of Applied Marine Biotechnology, Ministry of Education, Ningbo University, Ningbo, Zhejiang, China

## ABSTRACT

Marine mollusks are commonly subjected to heat stress. To evaluate the effects of heat stress on the physiological metabolism of the ark shell *Scapharca subcrenata*, clams were exposed to different high temperatures (24, 28 and 32 °C) for 72 h. The oxygen consumption and ammonia excretion rates were measured at 2, 12, 24, 48 and 72 h. The results indicated that the metabolic rates of the ark shell significantly increased with increasing heat stress, accompanied by mortalities in response to prolonged exposure. A metabolomics approach based on gas chromatography coupled with mass spectrometry was further applied to assess the changes of metabolites in the mantle of the ark shell at 32 °C. Moreover, multivariate and pathway analyses were conducted for the different metabolites. The results showed that the heat stress caused changes in energy metabolism, amino acid metabolism, osmotic regulation, carbohydrate metabolism and lipid metabolism through different metabolic pathways. These results are consistent with the significant changes of oxygen consumption rate and ammonia excretion rate. The present study contributes to the understanding of the impacts of heat stress on intertidal bivalves and elucidates the relationship between individual-level responses and underlying molecular metabolic dynamics.

## INTRODUCTION

Temperature is one of the more prominent abiotic factors that influences the physiological metabolism of animals and determines the ecological niche of a species (*Pörtner et al., 2006*; *Pörtner, 2010*; *Ezgeta-Balić et al., 2011*). Temperature has also become an increasingly severe source of environmental stress due to the increase of seawater temperature as a result of global climate changes. The global water surface temperature has increased by approximately 0.7 °C during the last century (*Hansen et al., 2006*) and a continued increase has been predicted (*Wang et al., 2015*). In response to environmental changes, organisms typically adjust their metabolic physiology to adapt to new energy requirements (*Cheung & Lam, 1995*; *Lagerspetz, 2006*; *Zhang et al., 2017*). It has been

Corresponding author
Baojun Tang, bjtang@yeah.net

reported that temperature can affect the metabolic rates of marine invertebrates, thus influencing the energy available for growth (*González et al., 2002*). Intertidal bivalves frequently face extreme heat stress (*Han et al., 2013*), and form reliable models to investigate the adaptations to highly fluctuating environments (*Davenport & Davenport, 2005*; *Wang et al., 2015*). Thus, studying the underlying metabolic alterations can help to understand the physiological changes that happen in bivalves in response to thermal stress.

The effects of heat stress on the energy metabolism of marine bivalves have been widely studied in many species (*Sokolova et al., 2012*), such as *Mytilus galloprovincialis* (*Anestis et al., 2007*), the limpet *Cellana toreuma* (*Han et al., 2013*), *Mercenaria mercenaria* (*Ivanina et al., 2013*) and the eastern oyster *Crassostrea virginica* (*Casas et al., 2018*). Among the many indicators of physiological responses to thermal challenge, the respiration behavior and individual-level metabolic rates (especially oxygen consumption rate) have been widely used to assess the physiological state in response to stress tolerance or adaptation during exposure to heat stress (*Sobral & Widdows, 1997*; *Saucedo et al., 2004*; *Sarà et al., 2008*; *Dowd & Somero, 2013*; *Frisk, Steffensen & Skov, 2013*; *Wang et al., 2015*; *Casas et al., 2018*). Successful persistence or tolerance requires molecular adaptations to compensate for the impaired metabolism, triggered by changes of temperature (*Lim et al., 2016*). Moreover, investigating the correlation between individual-level responses and molecular changes is useful toward a better understanding of the responses and regulating mechanisms from an overall perspective. Recently, research has increasingly focused on molecular adaptation or tolerance of marine bivalves to heat stress, and new analytic techniques, such as transcriptomics (*Lim et al., 2016*; *Nie et al., 2017*; *Yang et al., 2017*; *Juárez et al., 2018*; *Zhang et al., 2019*) and metabolomics (*Ellis et al., 2014*; *Digilio et al., 2016*), were used. The mantle tissue of mollusks has multiple functions, which include ligament secretion and sensorial activities; moreover, this tissue is very responsive to external stimuli (*Artigaud et al., 2015*). The mantle tissue has been used for transcriptome, proteomic, or metabolomic analyses in many studies (*Artigaud et al., 2014*, *2015*; *Wei et al., 2015*).

Metabolomics refers to the systematic study of chemical processes that involve metabolites. By measuring the levels of endogenous low-molecular-weight metabolites, metabolomics can be used to identify biomarkers that are indicative of physiological responses of living samples to specific environmental or culture conditions (*Alfaro & Young, 2018*). Many analytical platforms have been used for metabolomics, including raman spectroscopy, infrared spectroscopy, nuclear magnetic resonance (NMR) and many mass spectrometry (MS) techniques, of which NMR and MS are the most widely applied analytical tools for their sufficient high throughput and resolution properties (*Young & Alfaro, 2018*). Gas chromatography coupled with mass spectrometry (GC/MS) is a well-established analytical method that can provide a comprehensive and systematic understanding of all metabolites in biological samples (*Tsugawa et al., 2011*; *Nguyen & Alfaro, 2019a*). GC/MS-based metabolomics has been widely applied to study the physiological responses of aquatic organisms to environmental stressors, including pathogen infection, water contaminants and aerial exposure, and many metabolites and

associated pathways have been successfully identified (*Guo et al., 2014*; *Ji et al., 2016*; *Chen et al., 2015*; *Nguyen et al., 2018a*, *2018c*; *2019*, *Alfaro, Nguyen & Mellow, 2019*).

The ark shell *Scapharca subcrenata* inhabits the muddy sediments of the shallow coasts of China, Japan and Korea (*Nakamura, 2005*) and is widely cultured and consumed as a popular foodstuff in China and Korea (*Jin, Ahn & Je, 2018*). Due to their large geographic distribution, ark shell populations are exposed to strongly differing thermal conditions such as diurnal temperature fluctuations and extreme high temperature during summer. For example, the water temperature of the *S. subcrenata* habitat Xiangshan Bay (China) varies between 6.7 °C and 33.0 °C (*You & Jiao, 2011*). However, little information is available on the physiological responses of *S. subcrenata* to such pronounced heat stress.

Understanding the response of organisms to heat stress requires an in-depth understanding of both their acute responses and their compensatory acclimatization responses to the elevated temperature (*Pörtner et al., 2006*). In the present study, the oxygen consumption and ammonia excretion rates of *S. subcrenata* in response to heat stress were measured. Furthermore, the metabolic profile in the mantle was characterized using GC/MS to identify biomarkers for the responses to heat stress.

## MATERIALS AND METHODS

### Animals and heat stress

Adult *S. subcrenata* individuals were collected in Xiangshan Bay, East China Sea in November by a fishing trawler and transported in buckets containing seawater to the laboratory at the seaside of the Bay. The water temperature at the sampling site was 19.6 °C. The clams were maintained in a glass aquarium at 20 ± 0.5 °C, provided with constant aeration. To minimize the effects of body size on metabolic responses to heat stress, only individuals with similar shell length were used (28.11 ± 1.36 mm).

The clams were randomly divided into four groups (70 clams for 20, 24 and 28 °C treatment, and 80 clams for 32 °C treatment) and were transferred to four 60 L water baths, filled with aerated seawater, that were connected to a temperature controller. The seawater temperature was gradually increased from 20 to 24, 28 and 32 °C over 2, 4 and 6 h periods, respectively. The temperature was then maintained constant for further 72 h. At 2, 12, 24, 48 and 72 h of exposure to different heat stress levels, both the oxygen consumption rate and ammonia excretion rate of *S. subcrenata* were measured. During the experiment, the seawater in each tank was renewed daily.

Based on the results of oxygen consumption and ammonia excretion measurements, six replicates of mantle samples were taken 2 h and 24 h after the seawater temperature was gradually increased from 20 °C to 32 °C. All samples were immediately frozen in liquid nitrogen and stored at −80 °C for further metabolomic analysis.

### Measurement of oxygen consumption rate and ammonia excretion rate

Both the oxygen consumption and ammonia excretion rates were determined in a 1,500 mL glass respiration chamber. Two clams were sealed for 2 h in a chamber filled with oxygen saturated seawater. The oxygen concentration in this chamber was measured according to standard procedure (*Stickland & Parsons, 1968*). The $NH_4^+$-N concentration

was determined with the phenol-hypochlorite method (*Solorzano, 1969*). Individual clams were sampled for measurement at 1, 11, 23, 47 and 71 h after heat stress challenge. Since each measurement cycle spanned a 2 h period, the measured values represent the averages of 1–3, 11–13, 23–25, 47–49 and 71-73 h, respectively. The results are presented as the rates at 2, 12, 24, 48 and 72 h. Each measurement was performed in three replicates and a chamber without clams served as control. After measurements, the dry weight of the soft parts of each clam was determined after drying at 65 °C for 24 h.

## Metabolite extraction

For metabolite extraction, 30 mg accurately weighed wet sample was transferred into a 1.5 mL Eppendorf tube with two small steel balls (with diameters of 1.50 mm) for crushing. An aliquot of 20 µl of 2-chloro-l-phenylalanine (0.3 mg/mL), dissolved in methanol as internal standard and 600 µl mixture of methanol and water (4/1, vol/vol) were added to each sample. All samples were cooled to −80 °C for 2 min and then ground at 60 Hz for 2 min. After vortexing, the ground samples were ultrasonicated for 10 min at ambient temperature and cooled to −20 °C for 30 min. The samples were then centrifuged at 13,000 rpm, 4 °C for 15 min (Eppendorf Centrifuge 5427 R; Hamburg, Germany). The supernatant (400 µl) was dried in a freeze concentration centrifugal dryer (Christ RVC 2-33IR; Osterode, Germany), and 80 µl of 15 mg/mL methoxylamine hydrochloride in pyridine was added subsequently. The resulting mixture was vortexed for 2 min and incubated at 37 °C for 90 min. Then, 80 µl of bis(trimethylsilyl)trifluoroacetamide (BSTFA) (with 1% trimethylchlorosilane) and 20 µl n-hexane was added, which was vortexed for 2 min, and then derivatized at 70 °C for 60 min. The resultant mixture was exposed to ambient temperature for 30 min before GC-MS analysis.

Quality control (QC) samples were prepared by pooling all samples together, with the volume same as the analytic samples, and then analyzed using the same method. The QCs were injected at regular intervals (every 10 samples) throughout the analytical process to ensure the reproducibility of the GC-MS measurement.

## GC/MS analysis

The derivatized samples were analyzed using an Agilent 7890A GC system coupled with an Agilent 5975C MSD system (Agilent, Santa Clara, CA, USA). A HP-5MS fuzed-silica capillary column (30 m × 0.25 mm × 0.25 µm, Agilent) was utilized to separate the derivatives. Helium (>99.999%) was used as carrier gas at a constant flow rate of 6.0 mL/min through the column. The injector temperature was maintained at 280 °C and the injection volume was one µl in splitless mode. The initial oven temperature was 60 °C, was increased to 125 °C at a rate of 8 °C/min, to 190 °C at a rate of 10 °C/min, to 210 °C at a rate of 4 °C/min, to 310 °C at a rate of 20 °C/min, and finally, a temperature of 310 °C was sustained for 8.5 min. The temperatures of the MS quadrupole and ion source electron impact (EI) were set to 150 °C and 230 °C, respectively, and the collision energy was 70 eV. Mass data was acquired in full-scan mode (m/z 50–600), and the solvent delay time was set to 5 min.

## Statistical analysis

For the results of oxygen consumption rates and ammonia excretion rates, One-way ANOVA and multiple comparisons were performed with SPSS 11.5 statistical software, and a $P$ value less than 0.05 was considered to be statistically significant.

The acquired MS data from GC-MS were analyzed by ChromaTOF software (v4.34; LECO, St. Joseph, MI, USA). The metabolites were identified by the Fiehn database using the method described by *Mishra, Gong & Kelly (2017)*. Briefly, after alignment with the Statistic Compare component, the CSV file was obtained with three-dimensional data sets including sample information, peak name, retention time, m/z and peak intensities. The internal standard was used for QC. The internal standards and any identified pseudo positive peaks, such as peaks caused by noise, column bleed and the BSTFA derivatization procedure, were removed from the data set. Peaks from the same metabolites were combined.

The resulting data were normalized to the total peak area of each sample using Excel 2007 (Microsoft, Redmond, Washington D.C., USA) and were imported into the SIMCA software package (v14.0; Umetrics, Umeå, Sweden), where principal component analysis (PCA), partial least-squares discriminant analysis (PLS-DA), and orthogonal partial least-squares discriminant analysis (OPLS-DA) were performed. The Hotelling's T2 region, shown as an ellipse in score plots of the models, was used to define the 95% confidence interval of the modeled variation. The quality of the models was described by the $R^2X$ or $R^2Y$ and $Q^2$ values. $R^2X$ or $R^2Y$ are defined as the proportion of variance in the data that can be explained by the models and indicates goodness of fit. $Q^2$ is defined as the proportion of variance in the data that is, predicted by the model and indicates predictability, as calculated by a cross-validation procedure. A default seven-round cross-validation in SIMCA was performed to determine the optimal number of principal components and to avoid model overfitting. The OPLS-DA models were also validated by permutation analysis (200 times).

The different metabolites were selected based on the combination of a statistically significant threshold of variable influence on projection (VIP) values obtained from the OPLS-DA model and on $P$ values from a two-tailed Student's $t$ test on the normalized peak areas. Metabolites with VIP values exceeding 1.0 and $P$ values of less than 0.05 were included.

# RESULTS

## Survival rates

Enhanced heat stress increased the mortality of experimental clams. The survival rates of *S. subcrenata* exposed to 20, 24, 28 and 32 °C seawater temperature were 97.14%, 92.85%, 84.28% and 75%, respectively. Most mortalities occurred within 24 h after thermal challenge.

## Metabolic rates

After the temperature was increased to 24 °C, the oxygen consumption rate of *S. subcrenata* decreased for 2 h, followed by a significant increase at 24 h and 48 h ($P < 0.05$

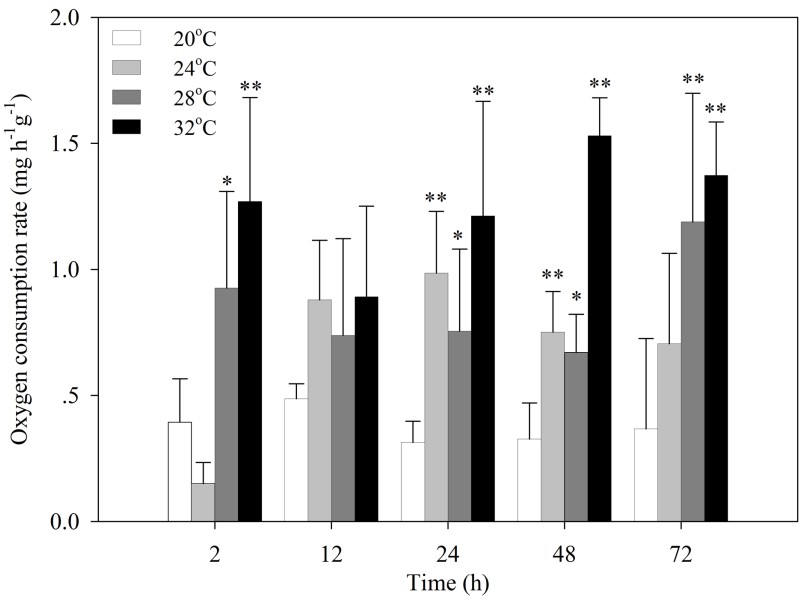

**Figure 1 Oxygen consumption rates of *S. subcrenata* after exposed to different temperatures.** Asterisks indicate significant differences (*$P < 0.05$; **$P < 0.01$) between the stressed and control group.

and $P < 0.01$, respectively; Fig. 1). After exposure to 28 °C and 32 °C stress, the oxygen consumption rates increased significantly at 2, 24, 48 and 72 h ($P < 0.05$, 0.01). The increase was not significant for 12 h, although the oxygen consumption rate was still higher than that of the control group.

After exposure to heat stress, the ammonia excretion rates of *S. subcrenata* increased first, then decreased, and increased again (Fig. 2). *S. subcrenata* exposed to heat stress at 24 °C demonstrated higher ammonia excretion at 12 h compared with the control group ($P < 0.01$). The clams exposed to 28 °C heat stress showed significantly higher ammonia excretion rates at 2, 12 and 72 h ($P < 0.05$ and $P < 0.01$, respectively). The ammonia excretion rate of *S. subcrenata* exposed to 32 °C was significantly higher than that of the control group at 2, 24, 48 and 72 h ($P < 0.01$).

## Metabolic profiles analyzed by GC-MS

The typical total ion chromatograms (TIC) of both *S. subcrenata* mantle samples and QC samples displayed stable retention times (Fig. S1). Thus, the TIC could directly reflect the differences of metabolite profiles among groups. The PCA score plot is shown in Fig. 3. The three groups were generally separated, especially the control group and the M-24 group. The $R^2X$ value of the PCA model, representing the explained variance for the groups, was 0.484. All groups were within the Hotelling ellipse of 95% confidence, indicating that the analyzed samples contained no outlier.

Further supervised pattern recognitions, PLS-DA and OPLS-DA, were performed to obtain a better explanation of different metabolic patterns (Fig. 4). The classification parameters for the PLS-DA and OPLS-DA model are shown in Table 1, indicating the robustness of the models. The pairwise groups in each subplot were clearly separated into

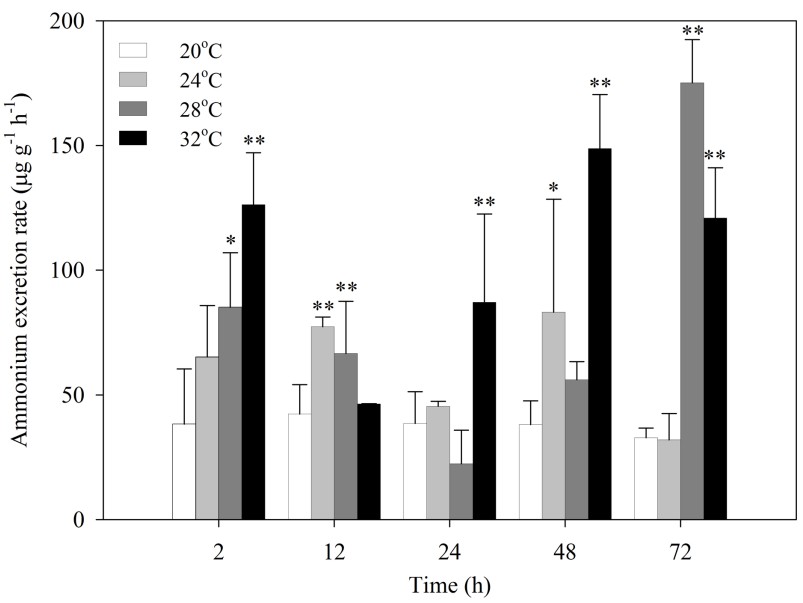

**Figure 2 Ammonia excretion rates of *S. subcrenata* after exposed to different temperatures.** Asterisks indicate significant differences (*$P < 0.05$; **$P < 0.01$) between the stressed and control group.

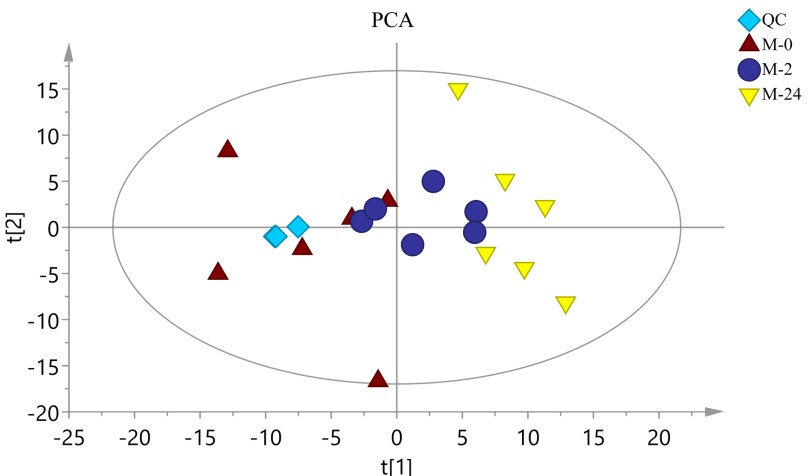

**Figure 3 The score plot of PCA.** QC is the quality control sample. M-0 is the control group, and M-2 and M-24 are the samples taken at 2 h and 24 h, respectively. The Hotelling ellipse indicating 95% Confidence Interval.

two sides of the Hotelling T2 ellipse, indicating that both models could identify differences among groups.

## Metabolite identification and comparison

A total of 345 metabolites were identified, including amino acids (e.g., aspartic acid, glutamic acid and histidine), organic acids (e.g., creatine, gluconic acid, malic acid and oxalic acid), and energy metabolism-related metabolites (e.g., glucose-6-phosphate and erythrose).

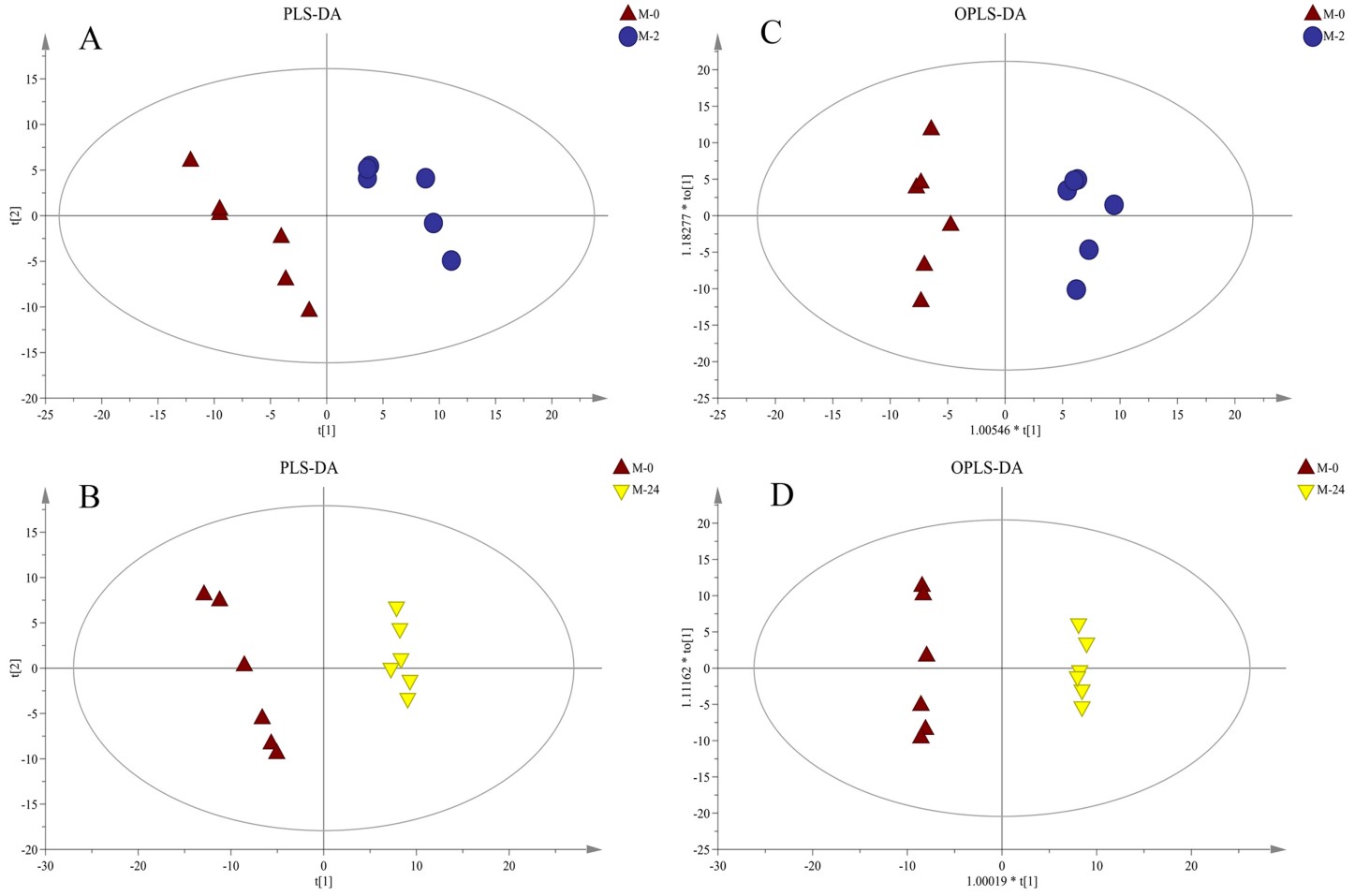

**Figure 4 PLS-DA and OPLS-DA score plots derived from metabolite profiles of *S. subcrenata*.** M-0 is the control group, and M-2 and M-24 are the samples taken at 2 h and 24 h, respectively. (A) PLS-DA score plot of the M-0 group and M-2 group, (B) PLS-DA score plot of the M-0 group and M-24 group, (C) OPLS-DA score plot of the M-0 group and M-2 group, (D) OPLS-DA score plot of the M-0 group and M-24 group.

**Table 1 Multivariate analysis of metabolite profiles of *S. subcrenata* after exposure to heat stress.** Two principal components for the PCA model, two principal components for the PLS-DA model, and one principal component and one orthogonal component for the OPLS-DA model were used.

| Model type | 0–2 h | | | | | 0–24 h | | | | |
|---|---|---|---|---|---|---|---|---|---|---|
| | $R^2X$ (cum) | $R^2Y$ (cum) | $Q^2$ (cum) | $R^2$ | $Q^2$ | $R^2X$ (cum) | $R^2Y$ (cum) | $Q^2$ (cum) | $R^2$ | $Q^2$ |
| PLS-DA | 0.3 | 0.97 | 0.445 | | | 0.36 | 0.991 | 0.841 | | |
| OPLS-DA | 0.3 | 0.97 | 0.381 | 0.963 | −0.033 | 0.468 | 0.999 | 0.833 | 0.993 | −0.003 |

At 2 h after heat stress, 39 metabolites showed significant changes, including 13 downregulated and 26 upregulated metabolites (Table S1). At 24 h, 90 metabolites showed significant changes, including 39 downregulated and 51 upregulated (Table S2).

Among metabolites that showed significant changes, the relative concentration of glucose-6-phosphate presented a continuous decrease at 2 h and 24 h, in contrast to the continuous increase of o-phosphorylethanolamine and taurine (Fig. 5). Latic acid was

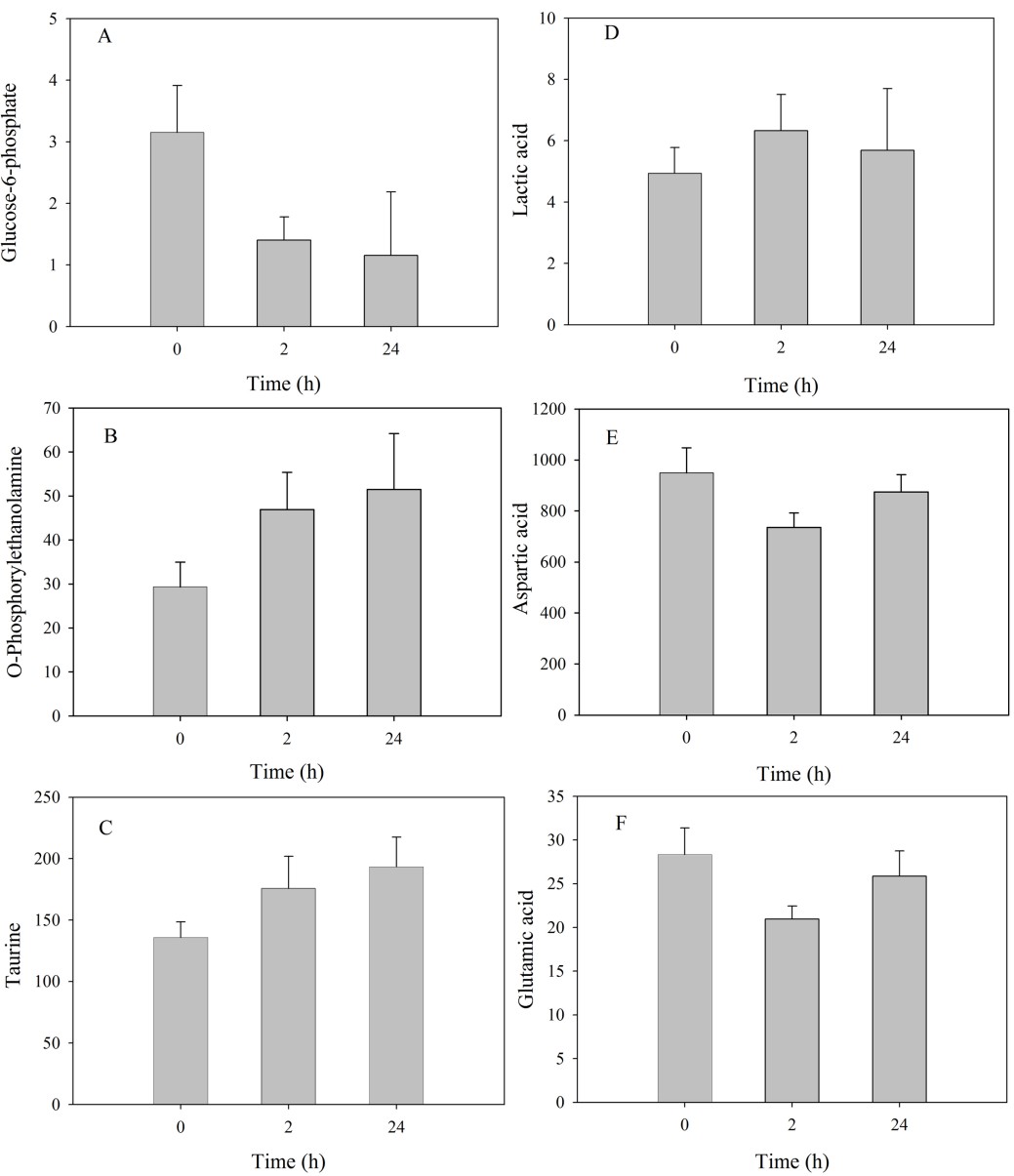

**Figure 5 Metabolite levels of metabolites in the mantle of *S. subcrenata* after exposure to 32 °C.**
(A) Glucose-6-phosphate, (B) O-phosphorylethanolamine, (C) taurine, (D) lactic acid, (E) aspartic acid and (F) glutamic acid.

significantly upregulated at 2 h but decreased to the control level at 24 h; aspartic acid and glutamic acid were significantly downregulated at 2 h but increased to the control level at 24 h.

## Metabolic pathway of common metabolites

KEGG pathway analysis was performed using MetaboAnalyst 3.0 software. Metabolic pathways were identified against the *Danio rerio* KEGG (zebrafish) library. Pathway topology analysis was performed based on the relative betweenness to calculate the important values. The metabolites were mapped onto 15 pathways for samples taken at 2 h

**Table 2 Pathway enrichment analysis of the metabolites at 2 h after exposed to 32 °C.**

| Pathway | $P$ | $-\log(P)$ | Impact |
|---|---|---|---|
| Histidine metabolism | 0.026555 | 3.62 | 0.0 |
| Alanine, aspartate and glutamate metabolism | 0.071896 | 2.63 | 0.45253 |
| D-Glutamine and D-glutamate metabolism | 0.090061 | 2.41 | 1.0 |
| Taurine and hypotaurine metabolism | 0.12386 | 2.09 | 0.2 |
| Nitrogen metabolism | 0.15645 | 1.86 | 0.0 |
| Arginine and proline metabolism | 0.19022 | 1.66 | 0.09612 |
| Nicotinate and nicotinamide metabolism | 0.23292 | 1.46 | 0.0 |
| beta-Alanine metabolism | 0.2616 | 1.34 | 0.0 |
| Sphingolipid metabolism | 0.32888 | 1.11 | 0.01504 |
| Pyruvate metabolism | 0.34161 | 1.07 | 0.0 |
| Butanoate metabolism | 0.34161 | 1.07 | 0.0 |
| Aminoacyl-tRNA biosynthesis | 0.35866 | 1.0254 | 0.0 |
| Glycolysis or Gluconeogenesis | 0.39025 | 0.94097 | 0.0 |
| Glutathione metabolism | 0.39025 | 0.94097 | 0.02968 |
| Porphyrin and chlorophyl metabolism | 0.40186 | 0.91166 | 0.0 |

**Table 3 Pathway enrichment analysis of the metabolites at 24 h after exposed to 32 °C.**

| Pathway | $P$ | $-\log(P)$ | Impact |
|---|---|---|---|
| Glyoxylate and dicarboxylate metabolism | 0.18143 | 1.7069 | 0.03704 |
| Synthesis and degradation of ketone bodies | 0.19888 | 1.615 | 0.0 |
| Cyanoamino acid metabolism | 0.23371 | 1.4537 | 0.0 |
| Butanoate metabolism | 0.2462 | 1.4016 | 0.0 |
| Taurine and hypotaurine metabolism | 0.26706 | 1.3203 | 0.2 |
| Biosynthesis of unsaturated fatty acids | 0.2724 | 1.3005 | 0.0 |
| Alanine, aspartate and glutamate metabolism | 0.27904 | 1.2764 | 0.0 |
| Nitrogen metabolism | 0.32952 | 1.1101 | 0.0 |
| Riboflavin metabolism | 0.38674 | 0.94999 | 0.16667 |
| Glycine, serine and threonine metabolism | 0.39216 | 0.93608 | 0.0 |
| Valine, leucine and isoleucine biosynthesis | 0.43916 | 0.82289 | 0.0 |
| Histidine metabolism | 0.46369 | 0.76853 | 0.2381 |
| beta-Alanine metabolism | 0.50963 | 0.67406 | 0.0 |
| Selenoamino acid metabolism | 0.53113 | 0.63275 | 0.0 |
| Pyrimidine metabolism | 0.53865 | 0.61869 | 0.08825 |

(Table 2; Fig. S2). Of these, the histidine metabolism pathway was significantly affected by thermal stress ($P < 0.05$). The D-glutamine metabolism, alanine, aspartate and glutamate metabolism exhibited higher impact values and was also affected even though no significant difference was observed. A total of 15 pathways were obtained when metabolites were imported into KEGG for samples taken at 24 h, and no pathway appeared to be significantly affected (Table 3). Interestingly, five pathways (glutathione metabolism, histidine metabolism, beta-alanine metabolism, nitrogen metabolism, alanine, aspartate

and glutamate metabolism, aminoacyl-tRNA biosynthesis) showed significant changes between 2h and 24h (Table S3).

## DISCUSSION

### Energy metabolism

The present study found a positive correlation between the oxygen consumption rate of *S. subcrenata* and the temperature of the surrounding environment. An increase in environmental temperature will result in the elevation of physiological rates and biochemical reactions, such as activities of mitochondria and metabolic enzymes, and other oxygen- and energy-demanding processes (*Ivanina et al., 2013*). The significant increase of oxygen consumption rate indicated the upregulation of both aerobic metabolism and energy demand (*Morley et al., 2012*). Many studies have indicated that the oxygen consumption rates of organisms typically increase with temperature until reaching a threshold temperature (*Shumway, 1982*; *Yukihira, Lucas & Klumpp, 2000*). Beyond this threshold, physiological rates can drastically decrease and anaerobic metabolic end products accumulate (*Sommer, Klein & Pörtner, 1997*; *Zhang et al., 2004*). This threshold is often referred to as the Arrhenius break temperature (ABT) (*Jansen, Hummel & Bonga, 2009*). In the present study, the oxygen consumption rate at 32 °C was always higher than at other temperatures, suggesting that the ABT for *S. subcrenata* is higher than 32 °C. However, the higher mortality (25%) at 32 °C suggests that such elevated heat stress might exceed the self-regulation capacity of this species despite the 6 h acclimation the clams experienced.

Variations in energy loss through respiration due to heat stress have been confirmed to influence the energy balance (*González et al., 2002*). Ammonia production is a result of the deamination of amino acids, and was also used to evaluate the energy loss of organisms when faced with environmental stress (*Wang et al., 2011*; *Shin, Chan & Cheung, 2014*). Evidence showed that amino acids may be catabolized after being released from cells, resulting in the increase of blood ammonia concentrations and external ammonia excretion rate (*Pierce, 1982*; *Vitale & Friedl, 1984*). Thus, the rate of ammonia-nitrogen excretion reflects the rate of protein catabolism (*Widdows, 1978*). The increased ammonia excretion rate in the present study indicated that amino acids might be catabolized, suggesting increased energy demand during heat stress.

The metabolomics approach, coupled with multivariate analysis, allowed the successful investigation of metabolic changes in response to environmental stress (*Cappello et al., 2013*). Multivariate analyses identified a clear separation between the control and heat stressed groups, suggesting the existence of significant metabolic differences in the metabolic profile. Metabolic profiling and functional analysis of key metabolic pathways provided an overview of the metabolic status both before and after heat stress (*Digilio et al., 2016*; *Hao et al., 2018*). The present data indicated that higher temperatures caused a wide array of changes in the metabolite profiles of *S. subcrenata*. The metabolites with significant changes and their pathways differed greatly between clams that were exposed to heat-stress for 2 h and 24 h, which could be due to changes of metabolic substrates with prolonged stress exposure.

## Amino acid metabolism

Many studies have highlighted that heat stress could influence both the energy balance and energy homeostasis of aquatic invertebrates (*Sokolova et al., 2012*; *Han et al., 2013*). As shown by the pathway enrichment analysis, most metabolites that showed significant changes participated in amino acid metabolism (alanine, aspartate, glutamate, taurine, histidine and beta-alanine). Free amino acids account for a large fraction of the metabolome of marine invertebrates (*Cappello et al., 2013*) and can be oxidized to supply energy in the Krebs cycle. When the oyster (*Crassostrea sikamea*) was exposed to metal pollution, amino acids including threonine, alanine, arginine, glutamate, beta-alanine, aspartate and glycine decreased significantly (*Ji et al., 2016*). Exposure to $Cu^{2+}$ could result in alterations of 25 metabolites involved in oxidative stress responses and apoptosis processes in mussel *Perna canaliculus* (*Nguyen et al., 2018a*). In the present study, the relative concentrations of both aspartic acid and glutamic acid were significantly downregulated 2 h after heat stress, suggesting that these amino acids were possibly oxidized. This downregulation is consistent with the significant increase of the oxygen consumption rate. The oxidation of amino acids for energy expenditure is typically achieved via deamination (*McVeigh et al., 2006*), which would also explain the observed significant changes of the ammonia excretion rate at 32 °C. Taurine is also an osmolyte and plays an important role in osmotic regulation (*Preston, 1993*; *Cappello et al., 2013*). Elevation of taurine levels indicates a disorder in osmotic regulation of *S. subcrenata* under heat stress and similar responses have been reported for the abalone *Haliotis diversicolor* (*Lu et al., 2016*).

In addition to the elevated aerobic metabolism, changes of anaerobic metabolites involved in the energy metabolism were also observed in the present study. Glucose-6-phosphate lies at the start of two major metabolic pathways: the glycolysis pathway and the pentose phosphate pathway. Significant depletion of glucose-6-phosphate and activation of the glycolysis pathway suggested that heat stress led to an enhancement of the anaerobic metabolism of *S. subcrenata*. Moreover, the accumulation of lactic acid and Krebs cycle intermediate (pyruvic acid) suggested that the Krebs cycle was disrupted by a switch towards anaerobic respiration. A similar result was also found in mussel *P. canaliculus* infected *Vibrio* sp (*Nguyen et al., 2018b*; *Nguyen & Alfaro, 2019b*) and surf clam *Crassula aequilatera* exposed to thermal stress (*Alfaro, Nguyen & Mellow, 2019*). When the ambient temperature exceeded the ABT, anaerobic metabolism in the limpet *C. toreuma* was enhanced via the opine pathway to provide energy (*Han, Zhang & Dong, 2017*). In stonefly nymphs, accumulation of anaerobic metabolites (lactate, acetate and alanine) was observed when the animals reached critical temperature (*Verberk et al., 2013*). This transition to partial anaerobiosis at high temperature is likely a compensation for insufficient aerobic energy production and can be attributed to the limited capacity of oxygen uptake (*Sokolova et al., 2012*). In ectotherms, when the ratio of oxygen supply to oxygen demand decreases and shortages of oxygen arise, both the cardiac and respiration activities were insufficient to meet the elevated oxygen demand at higher temperature (*Frederich & Pörtner, 2000*; *Verberk et al., 2013*).

## Carbohydrate metabolism and lipid metabolism

In the present study, the significant variation of phospho-ethanolamine, fatty acids, cytosine and adenine indicated that the lipid and nucleotide metabolisms were also involved into responses to heat stress. Lipids play both functional and structural roles in biological processes such as for energy supply and the maintenance of biological membranes (*Lee, Park & Lee, 2018*). The oxidation of fatty acids produces acetyl CoA, which is an important intermediate metabolite of the Krebs cycle (*Roznere et al., 2014*). The significant decreases of azelaic acid, adipic acid, oleic acid, palmitic acid and pentadecanoic acid suggest that the clams used lipid energy reserves for the production of acetyl CoA. Phospho-ethanolamine is an intermediate of the phospholipid metabolism. Ethanolamine is part of phosphatidylethanolamine (PE), which forms the cytomembrane of animal cells (*McMaster, Tardi & Choy, 1992*). Ethanolamine is phosphorylated and enters the cytidine diphosphate pathway for PE synthesis (*Cheng et al., 2012*). It has been reported that variations in temperature could result in membrane lipid remodeling in the blue mussel *Mytilus edulis* and the oyster *C. virginica* (*Pernet et al., 2007*, *2008*). The present study found a significant upregulation of phospho-ethanolamine. This is inconsistent with the elevation of taurine, indicating that the membrane permeability might be influenced by exposure to heat stress. The intracellular concentrations of adenine nucleotides have been proposed as indicators of stress in aquatic organisms (*Vetter, Hwang & Hodson, 1986*). In the present study, the concentrations of both adenine and cytosine significantly increased 2 h after exposure to 32 °C, followed by increases of adenosine and cytidine-monophosphate at 24 h, indicating enhanced nucleotide synthesis. However, no significant change was observed in the adenosine 5′-monophosphate (AMP) levels. The thymine concentration was significantly increased at 24 h, while the thymidine concentration was significantly decreased, suggesting that the nucleotide metabolism during heat stress requires further investigation.

## CONCLUSIONS

In summary, the oxygen consumption and ammonia excretion rates were determined during stress caused by different elevated temperatures in *S. subcrenata*. A GC/MS-based metabolomics approach was applied to assess changes of metabolites at 32 °C. The results demonstrated that the clams increased their metabolic rates with elevated temperature, and mortality was observed under heat stress. Metabolite and functional analyses indicated that heat stress induced disturbances in energy metabolism, osmotic regulation, amino acid metabolism, carbohydrate metabolism and lipid metabolism via different metabolic pathways. The present study provides an important contribution to the understanding of the impacts of heat stress on the clam *S. subcrenata* and elucidates the relationship between whole-organism responses and molecular metabolic dynamics. To better understand the physiological response of bivalves to environmental stresses, the integration of different omics approaches, including transcriptomics, proteomics and metabolomics, may be adopted in the future research.

## ACKNOWLEDGEMENTS

We appreciate very much the assistance from Ms. Wenchao Liu and Mr. Peibo Bao in the lab measurement.

### Funding

This work was supported by the Special Fund for Agro-scientific Research in the Public Interest of China (201303047), the Natural Science Foundation of Shanghai (18ZR1450000), and the National Natural Science Foundation of China (41576167). The funders had no role in study design, data collection and analysis, decision to publish, or preparation of the manuscript.

### Grant Disclosures

The following grant information was disclosed by the authors:
Agro-scientific Research in the Public Interest of China: 201303047.
Natural Science Foundation of Shanghai: 18ZR1450000.
National Natural Science Foundation of China: 41576167.

### Competing Interests

The authors declare that they have no competing interests.

### Author Contributions

- Yazhou Jiang conceived and designed the experiments, performed the experiments, prepared figures and/or tables, authored or reviewed drafts of the paper, and approved the final draft.
- Haifeng Jiao performed the experiments, authored or reviewed drafts of the paper, and approved the final draft.
- Peng Sun prepared figures and/or tables, and approved the final draft.
- Fei Yin analyzed the data, authored or reviewed drafts of the paper, and approved the final draft.
- Baojun Tang conceived and designed the experiments, performed the experiments, analyzed the data, prepared figures and/or tables, authored or reviewed drafts of the paper, and approved the final draft.

### Data Availability

The raw measurements of metabolic rates, data matrix of metabolomics analysis, the metabolites with significant changes, pathway enrichment analysis of the metabolites between 2 h and 24 h, and all pathways enriched after exposure to 32 °C are available in the Supplemental Files.

### Supplemental Information

Supplemental information for this article can be found online at http://dx.doi.org/10.7717/peerj.8445#supplemental-information.

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
