# Peer review of "Metabolic response of Scapharca subcrenata to heat stress using GC/MS-based metabolomics"

_PeerJ, doi:10.7717/peerj.8445_

## Round 0.1 · original submission · Major Revisions

The reviewers have commented on your above paper. They indicated that it is not acceptable for publication in its present form.

However, if you feel that you can suitably address the reviewers' comments (included), I invite you to revise and resubmit your manuscript. Furthermore, the authors should to re-visit the pathway analysis components of this study in order to improve the manuscript.

·

Basic reporting

The manuscript is clear and well written in English with some minor grammar errors. However, the literature references are not sufficient which need to be developed more (please see details in the general comments). The discussion is not well organised.

Experimental design

The research question is clear and well defined and the experiment is well designed. However, there are some unclear points about the methods. It’s not clear about sample collection and preparation for metabolomics and there is no description of quality control of metabolomics analysis.

Validity of the findings

The study aimed to use GC-MS metabolomics to reveal insights into the impacts of heat stress on the clam S. subcrenata. The Data are well presented and results are novel and interesting that contribute to the understanding of temperature stress in marine invertebrates.

Additional comments

Line 64-65, please separate references for metabolomics and transcriptomics.
Line 74-77, it’s too general, could you please go more specific? Which are these environmental stressors? Pathogen infections, water contaminants, temperature? There are many GC-MS investigations for marine bivalves which have been published recently. A recent review paper by Nguyen and Alfaro (https://doi.org/10.1016/j.aquaculture.2019.734488) may help you to have an overview of the topic. In addition, a discussion on metabolomics studies on heat/temperature on marine mollucs/bivalves needs to be added. For example, a study by Alfaro and Nguyen on surf clam (https://doi.org/10.1016/j.aquaculture.2018.08.065). In general, this section needs to be expanded.
Is Young et al 2015 correct reference here for environmental stressor? Please check.
Line 77-79, I don’t think its correct for all environmental stressors. Again, you need to expand your literature review to see which environmental stressors have been studied with metabolomics approaches. Then, made a specific statement for the stressor and the organism that you are studying.
Line 92, please delete spectroscopy.
L109, how did you control the temperature when exchange water?
L111-114, should be mentioned in the introduction
L118, assay? It should be analysis or measurement. Please edit.
L133-134 do not match with lines 114-118. Please edit. Where did you get 30 mg weight samples?
L187, the mortality of what?
L132 and 147, Have did you do quality control for your analysis? Please see https://doi.org/10.1007/s11306-019-1556-8 as an example.
L188, 75.00%? please edit to make it consistent for the whole manuscript.
L217, “quantified” is this study used a quantitative method? How did you measure the concentration? I don’t see it in your method. Please check and edit.
L217-219, please reword, add “e.g.,” after (, such as…. including amino acids (e.g., aspartic acid,…),…
Discussion: No clear separation between paragraphs, so it a little difficult to follow.
L240-243, please check grammar.
L244-245, which studies? Need references here.
L289, Is it a new paragraph? Where does it end? Could you please re-organize this paragraph with a clear statement of your findings following relevant references to support your findings?
L280, please also re-consider to add https://doi.org/10.1039/c8mt00092a which is relevant to this one.
L295, it more relevant to cite references of bivalve studies that found an increase of lactic acid and TCA cycle under stress conditions such as:
https://doi.org/10.1007/s11306-019-1556-8
https://doi.org/10.1016/j.aquaculture.2018.08.065
https://doi.org/10.1016/j.jip.2018.08.008
In addition, more discussion about TCA cycle would be interesting since there are many recent studies in bivalves found the increases of TCA cycle intermediates.
Overall, based on your conclusions I suggest considering the following outlines for your discussion which a clear statement in each paragraph with supporting references.
• energy metabolism,
• osmotic regulation,
• amino acid metabolism,
• carbohydrate metabolism and lipid metabolism
Conclusion: May consider to add suggestions for future research?
Figure 5. What is the unit?

·

Basic reporting

The manuscript is written clearly with professional English used throughout, with the submission being ‘self-contained’ and representing an appropriate ‘unit of publication’.

The content of the introduction sufficiently and concisely describes the field background to demonstrate how the work fits into the broader field of knowledge and contains relevant sources of cited literature.

Overall the article is well-structured including figures and tables. Figures are relevant to the content of the article, of sufficient resolution, and appropriately described and labelled.

Raw data has been provided. I assume the metabolomics data provided has been pre-processed first (i.e., normalised to total peak area, internal standard, and biomass etc)? A clearer description of the GC-MS data provided would be useful.

Experimental design

The research is original primary research which fits within the aims and scope of PeerJ.

The research question is currently a bit ambiguous, although ‘omics studies do tend to be discovery-driven without prior hypotheses. This is OK in my opinion since the authors do state how their research fills an identified knowledge gap which is good.

The investigation appears to have been conducted rigorously and to a high technical standard – although further details on the quality of metabolite identifications are required to evaluate this properly. Please provide more information on how the metabolites were matched against the Fiehn database and what specific criteria were used. See my General Comments below.

Methods are mostly described with sufficient detail & information to replicate, but there are some areas which require more detail (e.g., metabolite identification process, statistical analyses criteria and reporting). See my General Comments below.

Validity of the findings

All underlying data have been provided; they appear to be robust (assuming the metabolite identifications are of high quality/standard).

However, the validity of the data and findings are currently difficult to evaluate, since the statistical methods need to be described in more detail and some justifications should be provided for particular method choices.See my General Comments below.

Additional comments

In this study, the authors provide a metabolomics-based analysis of clam mantle tissue after different levels of thermal stress treatment. The study provides new physiological information on how clams are able to adapt to thermal stress and could be a valuable contribution. While the manuscript is very well-written, there are a number of areas which require attention and clarification before a proper evaluation on the validity of the results and interpretation can be made. Most of these are centred around choices made for data processing and statistical analyses. Some specific comments follow, which could help to improve the manuscript.

Lines 133-134: Was the data normalised to sample-specific biomass at some point (or was each sample exactly 30mg)? This should be clarified.

Lines 133-134: Were the samples weighed frozen and wet? With biomass being based on wet weight? Please clarify.

Line 168: The authors normalize their data to the total peak area of the chromatograms. Authors should be aware that this technique assumes equivalent total metabolite signal per sample and is suitable only when the majority of all analytes remain constant. However, this ideal situation often does not hold due to the nature of the samples, therefore such normalization may distort data potentially masking true biological trends. It is more common to normalise the data to the internal standard, to account for systematic errors during metabolite extractions/derivatizations (e.g., pipetting variability, sample loss), potential derivative instability, and within- and between-batch variations (and preferably also using pooled QC samples). Please provide justification for the normalization approach used.

Lines 182-183: The selection criteria for metabolites were based on OPLS-DA VIPs >1.0 and t-test p-values <0.05. However, there is no mention of employing a method to control for multiple hypothesis testing during the univariate analyses. It is advised that this be controlled for using an appropriate technique (e.g., Bonferroni correction, False Discovery Rate threshold).

Line 161: Please provide more information on how the metabolites were matched against the Fiehn database and what specific criteria were used to do this. It is also advised that that the ‘Metabolite Identification Level’ be reported in the Methods section according to the recommendations of the ‘Metabolomics Standards Initiative’ (http://cosmos-fp7.eu/msi.html). This could be a good place to also state the ‘Metabolite Level’ certainty for the identifications.

Lines 168-179: Although the authors do perform proper model validations and report the relevant model statistics for their supervised multivariate analyses techniques, it appears they have not transformed and/or scaled their data prior to running these models. This is crucial for PCA/PLS-DA/OPLS-DA so that the data has normal probability density function, and all metabolites are given similar weighting during the analyses (see Berg et al. 2006 [DOI:10.1186/1471-2164-7-142]). Please provide justification for not transforming and/or scaling the data prior to PCA/PLS-DA and OPLS-DA.

Line 188: Survival was optimal at the highest temperature stress, and lowest at ambient temperature. These results seem counter intuitive. Some elaboration on these results would be useful in the discussion. Or has the list of survival rates been listed in the wrong order?... according to the discussion, this might be the case! Please double check this.

Lines 191-202: It is not clear which statistical analysis was conducted on these data. ANOVAs? Please clarify where appropriate.

Line 205: Here there is mention of QC samples. However, no mention of these were provided in the Methods section. Please include in the methods what these QC samples were and how they were prepared. Also, how were they used exactly for quality control purposes?

Methods Section: There is currently no information in the methods section for the pathway analysis that was performed. Please provide these details, and include which criteria were used (i.e., test statistic and measure of centrality and network topology, and which model reference pathway library was selected as a proxy for the marine mollusc focus in this study?). Which metabolites were entered into the quantitative enrichment analysis – all of them, or only the statistically significant ones? Also, the authors refer to the topology Impact Factor score as if has substantial relevance to the results, which it doesn’t – the authors should be interpreting these results based on the pathway analysis p-values in the first instance. The impact score is an indicator of how important the metabolites identified are to the network function only (is usually not changed by a stress treatment unless metabolites appear or disappear in one group or the other). A very high impact score can be obtained but if the p-value is also high then there is no treatment effect on the pathway. This should be reflected in the results and discussion when interpreting the outcome of this analysis. Note that metabolite transformation and/or scaling is usually performed prior to Pathway Analysis also.

Lines 229-235: This section requires some attention to better describe the results of the Pathway Analysis, with specific mention of pathways which were identified as being ‘statistically significant’.

Table 1: It is unclear how many components were used to obtain the R2(cum) and Q2(cum) values for these models. Please clarify. It is also not clear what the non-cumulative R2 and Q2 values represent for the OPLS-DA models – is this based on the permutation testing? Please clarify in the table caption.

Table 2: Only one pathway (histidine metabolism) shows signs that it may be affected (p<0.05) by thermal stress – but the impact score is zero which means that the metabolites identified are limitedly of importance to the pathway function. Consider this during interpretations. Note that it is good/transparent to also present the False Discovery Rate values for these pathways also, and the number of hits (i.e., x metabolites identified in the study which matched that particular KEGG reference pathway).

Table 3: No pathways appear to be significantly enriched according to their test statistics. Consider this during interpretations

Supplementary Tables 1 & 2: Please double check the fold change values. For example, 3-indoleacetonitrile in ST1 is currently listed as having a foldchange of 0.00. There are a number of similar entries in ST2 with highly significant p-values (P<0.0001) but foldchanges of zero which doesn’t seem right.

Discussion & Conclusion sections: These sections are generally well-written, and links the metabolite data with other works on marine invertebrate physiology. Although it seems that the pathway analysis does not provide much supporting evidence that numerous pathways were being affected, as stated in the conclusion

Other: I urge the authors to re-visit the pathway analysis components of this study. The authors have identified a very comprehensive list of metabolites, and I would have thought that many of the pathway impact values would typically have been higher because of this (?). Some suggestions in this area would be to 1) include all metabolites (significant and non-significant) in the analysis using their KEGG code IDs (can be time-consuming to obtain, but will best-ensure that good coverage across numerous pathways is obtained); 2) be sure to use an invertebrate pathway library (unfortunately there are no molluscan models and the researchers will need to select the most closely related taxa – perhaps C. elegans). Note that MetaboAnalyst v4.0 has recently updated their KEGG libraries for the Pathway Analysis modules.

---

## Round 0.2 · accepted · Accept

Thanks for editing the manuscript according to the given suggestions. I am pleased to confirm that your paper has been accepted for publication in PeerJ.

Thank you for submitting your work to this journal.

·

Basic reporting

The manuscript has significantly improved. The text is generally well written. I recommend it for publication.

Experimental design

Authors have added more details into the method section which made the manuscript is sufficient in details.

Validity of the findings

Data are well presented and results are novel and interesting that contribute to the understanding the effects of temperature stress on marine invertebrates.

Additional comments

Thanks for editing the manuscript according to my suggestions.